# Maternal Consumption of a Diet Rich in Maillard Reaction Products Accelerates Neurodevelopment in F1 and Sex-Dependently Affects Behavioral Phenotype in F2 Rat Offspring

**DOI:** 10.3390/foods8050168

**Published:** 2019-05-17

**Authors:** Melinda Csongová, Emese Renczés, Veronika Šarayová, Lucia Mihalovičová, Jakub Janko, Radana Gurecká, Antonio Dario Troise, Paola Vitaglione, Katarína Šebeková

**Affiliations:** 1Institute of Molecular Biomedicine, Medical Faculty, Comenius University, 81108 Bratislava, Slovakia; melinda.csongova@gmail.com (M.C.); domonkosemese@gmail.com (E.R.); v.sarayova@gmail.com (V.Š.); lucia.mihalovic@gmail.com (L.M.); jakubjanko7@gmail.com (J.J.); radana.kollarova@gmail.com (R.G.); 2Department of Biology, Faculty of Medicine, Slovak Medical University, 83303 Bratislava, Slovakia; 3Institute of Medical Physics, Biophysics, Informatics and Telemedicine, Faculty of Medicine, Comenius University, 81108 Bratislava, Slovakia; 4Department of Agricultural Sciences, University of Naples Federico II, 80055 Portici, Italy; antoniodario.troise2@unina.it (A.D.T.); paola.vitaglione@unina.it (P.V.)

**Keywords:** dietary advanced glycation end products, rat, pregnancy, lactation, second generation, physiological reflexes, locomotion, exploratory behavior, anxiety-like behavior

## Abstract

Thermal processing of foods at temperatures > 100 °C introduces considerable amounts of advanced glycation end-products (AGEs) into the diet. Maternal dietary exposure might affect the offspring early development and behavioral phenotype in later life. In a rat model, we examined the influence of maternal (F0) dietary challenge with AGEs-rich diet (AGE-RD) during puberty, pregnancy and lactation on early development, a manifestation of physiological reflexes, and behavioral phenotype of F1 and F2 offspring. Mean postnatal day of auditory conduit and eye opening, or incisor eruption was not affected by F0 diet significantly. F1 AGE-RD offspring outperformed their control counterparts in hind limb placing, in grasp tests and surface righting; grandsons of AGE-RD dams outperformed their control counterparts in hind limb placing and granddaughters in surface righting. In a Morris water maze, female AGE-RD F1 and F2 offspring presented better working memory compared with a control group of female offspring. Furthermore, male F2 AGE-RD offspring manifested anxiolysis-like behavior in a light dark test. Mean grooming time in response to sucrose splash did not differ between dietary groups. Our findings indicate that long-term maternal intake of AGE-RD intergenerationally and sex-specifically affects development and behavioral traits of offspring which have never come into direct contact with AGE-RD.

## 1. Introduction

Cooking is considered a pivotal step in human evolution [1,2]. Ancestral humans started cooking rapidly after gaining the ability to control fire and modern humans are biologically dependent on cooking. However, in recent decades we observe a shift from the preparation of meals by traditional culinary techniques towards a preference of calories-dense Western type diets (rich in fats, reducing sugars and salt), worldwide [3]. A Western diet is characterized by consumption of meals prepared under high temperature, precooked, highly-processed foods, fast-foods and snacks. These foods introduce important amounts of advanced glycation end-products (AGEs) into the diet.

AGEs are formed spontaneously in living organisms and in thermally processed or stored foods. In the classical Maillard reaction, they result from a non-enzymatic reaction between reducing sugars and free amino groups of amino acids, peptides, proteins, or DNA [4]. Alternatively, the formation of AGEs might be initiated by reactive oxoaldehydes, produced in several metabolic pathways [5]. Due to a plethora of potential metabolic pathways, AGEs represent a highly heterogeneous, complex and largely chemically uncharacterized group of compounds. AGE-modification affects the structure, and, thus, a function of proteins. Moreover, protein-bound AGEs interact with cell surface multiligand pattern-recognition receptor for AGEs (RAGE) [6,7,8]. AGE-RAGE interaction activates downstream signaling pathways, leading to the production of reactive oxygen species, cytokines, growth factors, and adhesive molecules, which finally promote oxidative stress, inflammation, and induce atherogenesis and diabetes [9,10]. Thus, accumulation of AGEs in biological fluids and tissues plays a pathogenetic role in the development of several diseases, (e.g., chronic kidney disease, diabetes mellitus, cardiovascular, immunological and neurological diseases, atherosclerosis, some types of cancer) and their complications [11,12,13,14,15,16,17].

In foods thermally processed by temperatures > 100 °C, such as grilling, baking, roasting or frying, AGEs are formed within minutes to hours. AGEs render foods attractive taste, odor and color [18]. The current concept of metabolic transit of dietary AGEs is based on experimental studies. Proteolytic degradation of AGE-modified proteins in the gastrointestinal tract is a prerequisite for potential absorption of free AGE-adducts (i.e., AGE-modified amino acids) and AGE-peptides. Some studies suggest that AGE-modified proteins, particularly if cross-linked, are less digestible [19,20]. However, it has been not confirmed in a different study [21] and neither experimental nor clinical studies reported protein malnutrition or nutritional deficits after consumption of AGEs-rich diets. AGE adducts cross intestinal barrier by simple diffusion [22], while transit of some glycated dipeptides is facilitated via intestinal dipeptide transporter-1 (PEPT1) [23]. In clinical and experimental studies, administration of AGEs-rich diet is generally associated with a rise in circulating AGE levels, or with increased urinary excretion [24,25,26,27]. Experimental data suggest that absorbed dietary AGEs are partially trapped (to a varying degree) virtually in all tissues [24,27,28,29]. 

Prenatal, perinatal, as well as early postnatal exposure to stressors might increase the susceptibility of offspring to manifest chronic degenerative diseases in later life [30]. This phenomenon is attributed to epigenetic modifications, central to developmental or phenotypic plasticity [31,32]. Global rocketing incidence of obesity, even in adolescents and young adults [33,34], prompted the research on the relationship between maternal over-nutrition and health outcomes of offspring. In humans, obesity per se might associate with impaired cognitive performance and memory [35]. Maternal obesity or excess gestational weight gain increase, among others, the risk of offspring to manifest neurodevelopmental, neurological and psychiatric disorders [36,37,38,39]. Experimental studies documented that maternal obesity (induced preconceptionally, during pregnancy and lactation via consumption of Western diets, e.g., high-fat, high-fat high-sugar, or cafeteria diet) might impair early physical and neurobehavioral development of offspring, and memory or behavioral traits in later life, independently from the diet consumed by progeny [40,41,42,43,44,45,46,47]. Importantly, if Western type diets were not sufficient to induce obesity in dams or sows, the offspring presented better cognitive performance and memory compared with offspring of those fed control chow [48,49,50]. Data on locomotion, anxiety-like and exploratory behavior are contradictory [50,51,52]. Only two studies explored the intergenerational effects of maternal (high-fat) diet on behavior and memory of progeny [53,54]. On the other hand, intergenerational effects of AGE-rich diet (AGE-RD) consumption have been completely ignored until now. 

The current study was designed to test the impact of maternal (F0 generation) challenge with AGE-RD on subsequent generations (F1, F2) of male and female rats consuming a standard chow. We hypothesized that exposure of F0 dams would result in intergenerational effects on early physical development, a manifestation of physiological reflexes, and later neurobehavioral phenotype. 

## 2. Materials and Methods

### 2.1. Animals

Weanling Wistar rats (16 females (F), 16 males (M)) were purchased from Anlab (Prague, Czech Republic). Males and females were caged separately, under controlled conditions (temperature 22 ± 2 °C, humidity 55 ± 10%, 12 h:12 h light-dark cycle), with *ad libitum* access to chow and water. The experiment had been approved by The State Veterinary and Food Administration of the Slovak Republic and was conducted in accordance with the EU Directive 2010/63/EU and Slovak legislation.

### 2.2. Diets

Standard rodent feed (Ssniff R/M-H, Spezialdiäten GmbH, Soest, Germany; major nutrients: Crude protein—19.0%, crude fat—3.3%, sugar—4.7%, starch—36.5%, nitrogen free extracts—54.1%, dry matter—87.7%; metabolizable energy—12.8 MJ/kg) was administered as a control chow (CTRL). AGE-RD was prepared by heating of Ssniff R/M-H at 120 °C for 30 min, which does not affect the nutrient composition of the diet [29,55]. Thus, both dietary formulas were nutritionally equivalent but differed in AGE content.

#### Quantification of Maillard Reaction Products in Feeds 

In both feeds, the content of chemically defined AGEs (e.g., *N*^ε^-(carboxymethyl) lysine-CML, *N*^ε^-(carboxyethyl) lysine-CEL, and furosine) was determined according to Troise et al. [56].

Chemicals: Acetonitrile, methanol and water for AGEs analysis were obtained from Merck (Darmstadt, Germany). The ion pairing agent perfluoropentanoic acid (nonafluoropentanoic acid, NFPA), formic acid, hydrochloric acid (37%), ammonium formate, the analytical standards (4,4,5,5-*d*_4_)-L-lysine hydrochloride (*d*_4_-Lys) and lysine analytical standards were purchased from Sigma-Aldrich (St. Louis, MO, USA). Analytical standards *N*^ε^-(2-furoylmethyl)-L-lysine (furosine), its labelled standard *N*^ε^-(2-furoyl(^2^H_4_)methyl)-L-lysine (*d*_4_-furosine), *N*^ε^-(carboxymethyl)-L-lysine (CML) and its deuterated standard *N*^ε^-(carboxy(^2^H_2_)methyl)-L-lysine (*d*_2_-CML) were obtained from Polypeptide laboratories (Strasbourg, France). *N*^ε^-(carboxyethyl)-L-lysine (CEL), its deuterated standard *N*^ε^-(carboxy(^2^H_4_)ethyl)-L-lysine (*d*_4_-CEL) were obtained from TRC (Toronto, ON, Canada).

Method: Briefly, rodent feed was hydrolyzed for 20 h at 110 °C in hydrochloric acid (6 M), filtrated by polyvinylidene fluoride filters (PVDF, 0.22 µm Millipore, Billerica, MA, USA), then 200 µL was dried under nitrogen flow. Samples were dissolved in 200 µL of water previously spiked with the internal standard mix (*d*_4_-lysine, *d*_4_-furosine, *d*_2_-CML and *d*_4_-CEL, final concentration 200 ng/mg of the sample). Samples were loaded and eluted onto polymeric Oasis HLB 30 mg cartridges (Waters, Wexford, Ireland) according to the procedure described by Troise et al. [56] Upon drying, AGEs were dissolved in 200 µL of water and 5 µL were injected. A Kinetex C18 column (2.6 µm, 100 mm × 2.1 mm, Phenomenex, Torrance, CA, USA) and the following mobile phases: A, 5 mM NFPA and B, acetonitrile 5 mM NFPA were used for separation of furosine, CML, CEL, lysine and their respective internal standards. Positive electrospray ionization was used for detection and the source parameters were selected as follows: Spray voltage 5.0 kV; capillary temperature 350 °C, dwell time 100 ms. The chromatographic profile was recorded in multiple reaction monitoring mode (MRM) by using an API 3000 triple quadrupole (ABSciex, Carlsbad, CA, USA). Mass spectrometry transitions, as well as analytical performances were monitored according to previously reported procedures [56]. The concentration of the four markers was reported as mg/100g of proteins, while lysine Amadori compound concentration was obtained by considering a conversion factor of 3.1 [57]. 

### 2.3. Experimental Design

The experimental design of the present research is illustrated in Scheme 1. At the age of five weeks old, females were divided into two groups of approximately equal body weight. Eight females were exposed to isocaloric AGE-RD for a total of ten weeks (four weeks before conception, three weeks during pregnancy, three weeks during lactation). Remaining F0 animals consumed standard chow throughout the experiment. Eight-weeks-old F0 females were individually paired in separate cages with F0 males to produce F1 offspring. The presence of sperm in vaginal smear was considered as gestation day zero (GD0). The pregnant and nursing dams were housed in individual cages and kept in conditions with minimal stress factors to avoid influencing the offspring development and behavior. At post-natal day (PND) 1, litters size was adjusted to 6–8. Two females and two males were marked and subjected to developmental observations and behavioral studies. After weaning, animals were group-housed with littermates of the same sex and fed standard chow. The F2 generation was produced from animals not subjected to above mentioned testing, by the mating of F1 females from CTRL or AGE-RD dams with CTRL males of other kin. Dams and their offspring (which number was reduced to 6-to-8) were maintained on a control chow for the duration of the study. 

### 2.4. Maternal Studies (F0)

#### 2.4.1. Body Weight and Chow Consumption

Dams were weighed at the initiation of each dietary regimen, before pregnancy, weekly during pregnancy and lactation, and at the sacrifice. Food consumption was monitored by weighing, before pregnancy, during pregnancy, and lactation. Mean daily food consumption and intake of lysine Amadori compound, CML and CEL were calculated. 

#### 2.4.2. Behavioral Tests

Behavioral phenotyping of the animals with the focus on locomotor and exploratory activity, anxiety-like behavior, and memory was employed, using a battery of behavioral tests. Animal behavior was recorded using a dedicated camcorder and the files were analyzed using the image and video processing system EthoVision XT 10.1 (Noldus Information Technology, Wageningen, Netherlands). The open field and the novel object recognition tests were conducted in the PhenoTyper cages (Noldus Information Technology, Wageningen, Netherlands).

Open field test: The floor of the square shaped arena (45 × 45 cm) was virtually divided into a central (20 × 20 cm) and a border zone. Animals were placed individually into the cage and monitored for 5 min. Rodents tend to avoid the unprotected area and concentrate their ambulation near the walls. Thus, the time spent in the central zone was evaluated to assess anxiety-like behavior. Total distance moved and velocity were measured to assess locomotion [58].

Novel object recognition: This test was conducted immediately after the open field test, which served as habituation for the arena. Testing consisted of two trials. In the first trial, animals were exposed to two identical objects, and were allowed to freely explore them for 5 min. One hour later, in the second trial, one of the objects was replaced with a novel object. The novel object differed from the familiar object in size, shape, texture and color, but not in location [59]. To assess recognition memory, we measured the time spent exploring the novel and familiar object. The results are reported as exploration ratio, i.e., time exploring novel/(time exploring novel + familiar objects).

Light-dark test: The testing box (40 × 55 cm) consisted of a light chamber (illuminated with 250–300 lx) with an entry hole to a dark chamber covered by a lid. The light chamber served as the starting zone, and animals were allowed to freely explore both chambers for 5 min. Rodents tend to avoid open illuminated areas. To assess anxiety-like behavior, time spent in the light part of the box was evaluated as a percentage of the total time [58]. 

Elevated-plus maze: The apparatus consisting of two opposite open and two opposite closed arms crossed in the center zone and was elevated to a height of 60 cm above the floor. Animals were placed into the center zone and observed for 5 min. As an index of anti-anxiety-like behavior, the preference of the open arms over the closed arms was assessed [58], as follows:frequency of open-arm entries (%) = open arms/(open + closed arms) × 100(1)
time spent on open arms (%) = open arms/(open + closed arms) × 100(2)

Splash test: The test was conducted in the home cage. Rats were sprayed with a 10% sucrose solution on the dorsal coat, and grooming behavior was observed for 5 min. To assess anhedonia, discomfort and self-care behavior, latency time to start grooming and total time spent by grooming were recorded [60].

#### 2.4.3. Sacrifice

Rats were sacrificed by i.p. overdosing of anesthetics (ketamin 100 mg/kg, Narkamon inj, Bioveta, Czech Republic; xylazine 10 mg/kg, Xylariem inj, Riemser, Germany). Abdominal, retroperitoneal, and perigonadal fat pads were removed and weighted. Fat-to-body weight was calculated.

### 2.5. Offspring Studies

Offspring (F1, F2) of F0 dams consuming control diet is referred to as CTRL; those of F0 dams fed AGE-RD diet as F0/AGE-RD.

#### 2.5.1. Body Weight and Chow Consumption

Offspring weight was monitored daily till PND21, thereafter rats were weighed in puberty (P, 7–8 weeks old), as young adults (A, 6-month-olds) and before sacrifice (old—O, 12 months old). Chow consumption was assessed by weighing. 

#### 2.5.2. Maturation of Physical Features and Development of Neurological Reflexes

Evaluations had been done daily between 09:00 and 12:00, by same investigators. All tests were carried-out in the same order. To minimize the effects of maternal separation and handling during daily testing, the pups were maintained under identical conditions to those of maternal living. 

During the suckling period, auditory conduit opening, i.e., the opening of the internal auditory conduits of both ears; incisor eruption, i.e., the first visible and palpable cresting of the incisors; and eye opening, i.e., any visible break in the membranes covering either eye, were inspected daily [61]. The maturation age of each feature was defined as the day on which the feature was observed for the first time.

Maturation of the nervous system was examined daily, from PND2 to PND21. Offspring were tested for the appearance of specific neurological reflexes and the time to perform a certain task [61]. Each day, the presence or absence of reflex was recorded. 

Sensory reflexes: The ear twitch and the eyelid reflex were tested. Positive reflex was considered when contraction appeared after a gentle touch to ear or eye lid with a cotton swab.

Limb placing: Presence of reflex was classified if the animal placed its fore- or hind-paw up to cardboard when it had been stroked against the dorsal surface of the paw.

Limb grasp: Grasping or not grasping onto a thin rod by the fore- and hind-limbs touched, was recorded.

Auditory startle reflex: Presence or absence of a whole-body startle response to a loud clapping sound was recorded.

Negative geotaxis: The test was considered positive if the animal placed head down in the middle of the inclined polystyrene board turned around and climbed up the board with the forelimbs reaching the upper rim within 30 s. 

Gait test: The test was considered positive if the pup placed in the center of a circle of 10 cm diameter moved off the circle with both forelimbs within 30 s.

Surface righting reflex: This reflex was considered positive if the animal turned over from supine position to prone position on all four paws within 1 s. 

Air righting reflex: If the animal landed on four paws when it was dropped head down from a height of about 45 cm onto a soft surface, the reflex was considered positive.

#### 2.5.3. Behavioral Tests

Except for the tests described above (Section 2.4.2), a Morris water maze test was performed. Animals were tested in puberty, as young adults and at 12 months of age.

Morris water maze: The Morris water maze test was conducted in a dark plastic circular pool (height 60 cm, diameter 135 cm) filled with water (25 ± 1 °C). The arena was virtually divided into four quadrants, which were marked for orientation and spatial learning with intra-maze cues. A hidden platform was placed in one of the quadrants, one cm under the water surface. Testing was performed for five days. During the first four days, each animal was tested four times per day, while a different quadrant has been used as a starting position in each trial. Animals were allowed to swim and search for the platform up to 60 s and subsequently they were left stay on the platform for another 30 s. If the animal was unable to find the platform within 60 s, it was gently guided to it by the experimenter. For each day, the average latency time to find the platform in four trials was calculated. To assess working memory, the performance of the animals during the first four days was evaluated. On the fifth day, the probe trial was performed. The platform was removed from the water maze, and animals were monitored for 60 s. The quadrant opposite to the platform-quadrant was used as a starting position. To assess reference memory, time spent in the platform-quadrant during this probe trial was evaluated [62].

#### 2.5.4. Sacrifice

F1 and F2 offspring were euthanized at the age of 13 months, by overdosing of anesthetics. Body weight and that of excised fat pads were determined.

### 2.6. Statistical analysis

Normality of data distribution was checked using a D’Agostino test. In F0 generation, CTRL and AGE-RD groups were compared using unpaired two-sided Student’s t-test; while body weight was evaluated using repeated-measures analysis of variance (ANOVA). In offspring, body weight and chow consumption were evaluated employing the general linear model (GLM), with diet and sex entered as fixed factors. The repeated measures GLM was used to assess the effects of F0 diets (CTRL, AGE-RD), age (P, A, O) and generation (F1, F2) on dependent variables in an elevated plus maze, an open field, light dark, novel object recognition and splash tests. If there was an effect of diet and age interaction, a separate analysis in each generation had been performed. Within-day performance in the Morris water maze was evaluated using GLM with diet and generations as fixed factors. Bonferroni correction was used in all multiple comparisons. In neurodevelopmental studies, mean PND of incisors eruption, ear conduit and eye opening, the appearance of an eye lid and auditory startle reflexes in CTRL and AGE-RD offspring had been compared employing unpaired two-sided Student’s t-test. Proportions were compared using the Fisher’s exact test. Differences were declared to be statistically significant at *p* < 0.05. If not indicated differently, data are given as mean ± standard deviation (SD), or as proportions. Since the manifestation of early physiological reflexes showed intra-individual day by day fluctuation, precise estimation of PND of reflex manifestation was cumbersome. Thus, on a day-by-day basis (from PND2 to PND21), percentage of animals manifesting reflex was plotted, and the area under the curve (AUC) was calculated (by summation of trapezoids between the different time points). The 15% difference in AUC (between two dietary groups) was arbitrarily set as significant. Analyses were carried-out separately in females (F) and males (M). Statistical analyses were performed using GraphPad Prism 6.0 (GraphPad Software, La Jolla, CA, USA), or IBM SPSS v. 20 software (IBM, Armonk, NY, USA).

## 3. Results

### 3.1. AGEs Content in Feed

In the control diet, furosine content reached 25.8 ± 2.1 mg/100 g proteins, that of lysine-Amadori compound 79.9 ± 6.5 mg/100 g proteins, CML 4.1 ± 0.4 mg/100 g proteins, and content of CEL was 1.1 ± 0.1 mg/100 g proteins; while AGE-RD contained about 34.7 ± 1.6 mg furosine/100 g proteins, 107.4 ± 4.8 mg lysine-Amadori compound/100 g protein, 6.1 ± 0.6 mg CML/100 g proteins, and 2.9 ± 0.2 mg CEL/100 g proteins. Thus, in comparison with the control diet, AGE-RD contained about 1.3-fold more furosine, 1.3-fold more lysine Amadori compound, 1.5-fold higher amount of CML and 2.7-fold higher amount of CEL. CTRL feed contained 1.1% nutritionally blocked lysine, measured according to Nursten [63], and including CML and CEL along with furosine, while in AGE-RD it represented 1.9%.

### 3.2. Maternal Data

Body weight of dams gradually increased from the day of their allocation into respective dietary group till weaning (*p* < 0.001) and did not differ significantly between CTRL and AGE-RD groups in either time interval (Figure 1). At sacrifice, the relative weight of fat pads did not differ significantly between the groups (*p* = 0.125).

During pregnancy, AGE-RD dams consumed about 9% (*p* = 0.027) less chow compared with their counterparts on control feed, while mean daily consumption during lactation (CTRL: 30.5 ± 2.6 g/d; AGE-RD: 25.7 ± 1.3 g/d, *p* = 0.105) was similar. In CTRL dams, dietary consumption of lysine Amadori compound reached 1.5 ± 0.1 mg/d, CML consumption reached 78 ± 0.4 μg/d, and that of CEL was about 20 ± 1 μg/d. During lactation, dietary load increased to 4.6 ± 1.0 mg/d, 106 ± 9 μg/d, and 51 ± 4 μg/d, respectively. Thus, pregnant AGE-RD dams consumed about 22% more lysine Amadori compound, 35% more CML, and 2.5-fold more CEL (*p* < 0.001, all) than their CTRL counterparts, while during lactation mean difference was about + 14% (*p* = 0.198), +25% (*p* = 0.033), and +130% (*p* < 0.001), respectively.

All AGE-RD dams successfully delivered healthy offspring. Two out of eight F0 CTRL dams experienced embryonic/fetal loss during gestation (*p* = 0.450). Litter size (CTRL: 15.5 ± 0.6 vs. 13.1 ± 1.6, *p* = 0.219) or sex distribution (CTRL: 44 F/48M, AGE-RD: 35F/50M, *p* = 0.374) did not differ significantly between the groups. 

Both groups of dams performed similarly in an open field-, light-dark-, and a novel object recognition tests, as well as in an elevated plus maze (Appendix A). In a splash test, AGE-RD dams started grooming later (*p* = 0.008), but no significance in grooming time has been observed (Appendix A).

### 3.3. Offspring 

#### 3.3.1. Body Weight and Chow Consumption

Newborn F1 offspring showed similar body weights (Table 1). GLM indicated that at weaning offspring of AGE-RD dams were heavier compared with their counterparts born to CTRL dams, but the post-hoc tests failed to indicate significance (F: *p* = 0.329, M: *p* = 0.262). Seven weeks-old AGE-RD offspring presented lower body weight than CTRL rats, but daily chow consumption did not differ significantly between the groups (Table 1). Maternal diet showed no significant effect on body weight of 6- and 12-months old offspring, their mean weight gain throughout the study, and relative weight of fat pads at sacrifice. At 12 months, AGE-RD offspring consumed more chow compared with CTRL offspring, but significance had been reached only in females (F: *p* < 0.001; M: *p* = 0.692) (Table 1).

In F2 offspring, no significant impact of grandmaternal diet on body weight and feed consumption had been recorded in either time point (Table 2). 

In both generations of offspring, sex differences in body weight were manifested from 7th week of age (Table 1 and Table 2). 

#### 3.3.2. Effects of F0 Diet on Maturation of Physical Features and Development of Neurological Reflexes in F1 and F2 Offspring

The mean day of incisors eruption, auditory conduit and eye opening, appearance of eye lid and auditory startle reflexes, and gait test performance did not differ between CTRL and AGE-RD offspring in either generation (Appendix A). Minimal differences in AUC of ear twitch and air righting reflexes, and forelimb placing between AGE-RD and CTRL offspring of both sexes in either generation suggest no significant difference in their manifestation (Appendix A). AUC > 15% in AGE-RD offspring suggested that F1 females and F1 males outperformed their CTRL counterparts in hind limb placing, in forelimb and hindlimb grasp and surface righting; and F1 males in negative geotaxis test. In F2, grandsons of AGE-RD dams outperformed their CTRL counterparts in hind limb placing and granddaughters in surface righting (Table 3 and Appendix A).

#### 3.3.3. Effects of F0 Diet on Behavioral Phenotype of F1 and F2 Offspring 

Locomotor activity: In an open field test, distance travelled significantly decreased in females by ageing (GLM: F_(2, 82)_ = 45.1, *p* < 0.001). GLM did not indicate significant effect of age and diet (*p* = 0.450) or age, diet and generation interaction (*p* = 0.396), (Figure 2A,B). Similarly, velocity of ambulation decreased by age (GLM: F_(2, 82)_ = 44.9, *p* < 0.001), without significant effects of age and F0 diet (*p* = 0.454) or age, F0 diet and generation interaction (*p* = 0.397), (Appendix A). Ageing similarly affected the distance travelled and velocity in male offspring (GLM: F_(2, 85)_ = 103.5, *p* < 0.001; both), (Figure 2C,D and Appendix A). While the effect of age and diet interaction was not significant (*p* = 0.061, both), that of age, F0 diet and generation interaction on velocity and distance travelled reached *p* = 0.018. Effect of age and F0 diet interaction was significant only in F2 generation (F_(2, 47)_ = 26.4, *p* = 0.003). In pubertal age, AGE-RD males ran more slowly and a shorter distance (*p* = 0.015, both) compared with their CTRL counterparts. Moreover, while CTRL males showed lower ambulation both in adulthood and at old age (*p* < 0.01; both) compared with puberty, AGE-RD offspring ran shorter distance with lower speed only in old age (*p* = 0.018, both).

*Anxiety-like behavior:* By ageing, offspring spent longer time in the central zone of the open field test (F: F_(2, 90)_ = 9.4, *p* < 0.001; M: F_(1, 63)_ = 5.2, *p* = 0.017). However, F0 diet showed no significant effect (age and F0 diet interaction: F: *p* = 0.070, M: *p* = 0.242; age, F0 diet and generation interaction: F: *p* = 0.185, M: *p* = 0.565), (Figure 3A–D). 

With increasing age, both female (F_(2, 85)_ = 19.6, *p* < 0.001) and male (F_(2, 85)_ = 20.4, *p* < 0.001) offspring spent longer time in illuminated part of light-dark box. However, GLM did not indicate a significant effect of age and F0 diet (F: *p* = 0.092; M: *p* = 0.459) or age, F0 diet and generation interactions (F: *p* = 0.159; M: *p* = 0.486), (Appendix A). Light-to-dark transitions frequency increased with ageing, both in females (F_(2, 85)_ = 13.1, *p* < 0.001) and in males (F_(2, 76)_ = 21.0, *p* < 0.001), Table 4. Age and F0 diet (*p* = 0.219) and age, F0 diet and generation (*p* = 0.882) interactions were insignificant in females, while a significant interaction between age, F0 diet and generation (*p* = 0.042) was observed in males. Further analysis revealed that transition frequency was significantly lower in F2 AGE-RD offspring compared with their CTRL counterparts (*p* < 0.05).

3-way GLM indicated significance in time spent in open arms in the elevated plus maze test (F: F_(2, 221)_ = 20.5, *p* < 0.001; M: F_(2, 96)_ = 8.5, *p* < 0.001), (Figure 4A–D), as well as in open arms entries frequency (F: F_(2, 43)_ = 29.2, *p* < 0.001; M: F_(2, 38)_ = 14.3, *p* < 0.001), (Appendix A). However, F0 diet showed no significant impact in either setting (age and F0 diet interaction: *p* = 0.176, 0.770, 0.774 and 0.348, respectively). 

Exploration: In the novel object recognition test, repeated measures 3-way GLM indicated significant difference in a preference to sniff a novel object in females (GLM: F_(2, 100)_ = 6.3, *p* = 0.003), but significant effect of F0 diet has not been confirmed (age and F0 diet interaction: *p* = 0.300; age, F0 diet and generation interaction: *p* = 0.697). Neither of investigated factors affected significantly the exploration ratio in male offspring (GLM: F_(2, 76)_ = 1.3, *p* = 0.286), (Appendix A).

Learning and memory: In the Morris water maze, female F1 and F2 offspring from both maternal diet groups displayed an approximately similar ability to acquire the task on the first training day, as shown by the latency to find the platform (Figure 5). Both generations of female CTRL offspring appeared to have greater difficulty learning the task compared with AGE-RD offspring, as reflected by longer latencies to locate the platform on days 2-to-4 (repeated measures GML: F_(3, 121)_ = 62.7, *p* < 0.001; day and F0 diet interaction: *p* = 0.013; day, F0 diet and generation interaction: *p* = 0.395). On trial days 2-to-4, F1 AGE-RD females outperformed their CTRL counterparts. 

An ability to locate the platform during training sessions improved also in male offspring (GLM: F_(3, 121)_ = 57.2, *p* < 0.001), (Figure 6). In contrast to females, no significant effects of F0 diet were observed (day and F0 diet interaction: *p* = 0.639; day, F0 diet and generation interaction: *p* = 0.833).

When the ability to remember the location of the escape platform was tested in a probe test, i.e., one day following the last training trial, no difference was observed between F0 dietary groups of offspring of either generation (Table 5).

Splash test: No significant difference has been observed in latency to groom (F: F_(2, 99)_ = 1.6, *p* = 0.215; M: F_(2, 93)_ = 0.61, *p* = 0.427) or grooming time (F: F_(2, 89)_ = 0.83, *p* = 0.432; M: F_(2,91)_ = 0.60, *p* = 0.545), (Appendix A).

## 4. Discussion

We describe for the first time, sex-dependent intergenerational heritability of grandmaternal AGE-RD via the maternal line on the short-term memory and some features of anxiety-like behavior in rats. These effects were observed in the absence of maternal obesity, in offspring never coming into direct contact with AGE-RD.

In this study, F0 dams were exposed to AGE-RD preconceptionally, during gestation and lactation. F1 animals were therefore potentially exposed as germ cells, during intrauterine development and while suckling, whereas F2 animals only as germ cells of the F1. In mice, dietary AGEs accumulate in gonads [28,29], but it is uncertain in which type of cells. In poorly controlled diabetic rabbits, AGEs accumulate in the endometrium and the embryo [64], but whether accumulated AGEs are of endogenous or exogenous origin, remains unclear. It has not been documented yet, whether dietary AGEs cross the placental barrier. Based on the observation of a tight relationship between maternal plasma AGE levels at delivery and those of their newborns [65], maternal transfer of dietary AGEs is highly probable. 

In experimental studies, heating of commercially available feed to about 120 °C for approximately 30 min. is a common method to obtain AGE-RD [29,55]. Due to the difference in raw materials and composition of standard rodent diets, it is expectable that similar heat-processing will yield a different degree of AGEs modification. In the majority of studies, the content of AGEs in diets has been measured using unspecific ELISA method [66,67]: Epitopes reacting with the employed antibody are not limited to AGE-moieties, while the antibodies probably do not recognize all AGE epitopes [68,69]. The ELISA method quotes AGE content of feed in kU/weight, not allowing for comparison with amounts of chemically defined AGEs, as quantified in our study through ion pairing liquid chromatography tandem mass spectrometry and as outlined in dietary AGEs database [70].

Besides nutritionally blocked lysine and furosine, CML is a widely used marker to assess AGE-modification in foods [70,71]. In randomized clinical trials focusing on the effects of AGE-rich vs. AGE-poor diet intake, about 1.3-to-2.3-fold difference in CML intake had been reported [72,73,74]. In our study, 1.3- and 1.4-fold increase in CML intake during pregnancy and lactation was achieved, respectively. This might realistically correspond to the difference in dietary AGEs consumption between humans preferring foods prepared according to traditional culinary recipes and those preferring fast foods and snacks. 

Dams consuming AGE-RD over 10 weeks displayed similar body weight throughout the study and similar adiposity at sacrifice as their counterparts consuming a control diet. Thus, the effects of F0 maternal obesity on outcomes observed in offspring might be excluded.

In the present study, prenatal, gestational and lactational challenge of F0 dams with AGE-RD did not affect their reproductive ability and that of their F1 female offspring. F0 CTRL and AGE-RD dams showed similar locomotor activity, exploratory and anxiety-like behavior. In our former study, three-weeks long administration of AGE-RD (75% control feed: 25% bread crusts) to adult male rats did not affect significantly their locomotor or exploratory activity, but in light-dark box test, anxiogenic effects were observed [75]. It cannot be excluded that different AGE-RD does not exert the same effects, or that the effects of AGE-RD might be relieved after conversion to a standard diet. We did not test the dams during the period of AGE-RD intake.

Literature data on the effects of the maternal dietary challenge are scarce. Indeed, monitoring of early development and appearance of physiological reflexes in offspring is generally the domain of pharmacological and toxicological studies, or those on the effects of prenatal and/or perinatal stress imposed on dams [61]. In this study, F0 AGE-RD consumption did not influence significantly somatic development of offspring of the next generation. Data on the intergenerational effects of F0 maternal challenge with a Western type diet throughout gestation and lactation on somatic development of pups are not available. In F1 rats, maternal intake of a high-fat diet during pregnancy associated with a delay in auditory conduit opening and incisors eruption if compared with offspring of dams on a standard diet; while intervention restricted to lactation period resulted also in a delay of ear unfolding and eye opening [76]. In mice, maternal intake of cafeteria diet during pregnancy associated with a delayed incisor eruption, eye opening, and palmar grasp [46]. Delay in some features of physical maturation in offspring of dams consuming a Western type diet is probably independent of maternal obesity, as the delay was documented both in the absence [76], as well as in presence, of excessive maternal gestational weight gain [46]. In contrast to these data, but in accordance with the present experiment, our former study in mice fed AGE-RD (75% standard chow:25% bread crusts) during pregnancy showed that somatic development of offspring did not differ from that of control feed consuming dams [77]. Interestingly, perinatal exposure to diet sub-optimal in the provision of polyunsaturated fatty acids did not affect the manifestation of somatic features in F1 mice [78]. 

While in rodents maternal high-fat or cafeteria diets postpone manifestation of physiological reflexes in F1 offspring [46,76], AGE-RD in F0 generation associated with their earlier manifestation [77]. In the current study, an earlier manifestation of reflexes was less frequent in F2 than in F1 generation. It is worth noticing that the AGE-RD offspring performed better in surface righting. Adequate performance of this complex reflex requires integrity of muscular and motor function and adequate acquisition of symmetrical coordination between left and right sides of the body [79]. Dietary CML crosses the blood-brain barrier and accumulates within the brain [28]. The rise in AGEs content in the central nervous system is a feature of aging [80]. Therefore, the manifestation of physiological reflexes in earlier developmental stages might reflect earlier ageing. 

While F0 AGE-RD affected the manifestation of physiological reflexes mainly in F1 offspring, the neurobehavioral phenotype was affected more frequently in the F2 generation. F0 AGE-RD associated with lower locomotor activity in an open field test in F2 males. F2 AGE-RD males also exhibited a lower number of light-to-dark transitions in a light-dark test, which might be indicative of anxiolysis. However, in other tests, anxiolytic outcomes have not been confirmed. Different responses in different types of test do not allow a clear conclusion to what extent AGE-RD diet of grandmothers influences anxiety in F2 male offspring. Similarly, other studies documented that maternal dietary challenge diversely impacts offspring locomotion and anxiety. Indeed, offspring from dams consuming a cafeteria diet showed either lower locomotion and lower anxiety in an elevated plus maze and an open field test [44] or higher locomotion and higher anxiety in an open field test (but not in an elevated plus maze) [51] than control offspring. Maternal high fat diet intake had no significant impact on locomotor activity of adult F1 [42,43,81] or F2 [54] offspring; while in other studies, voluntary physical activity of F1 offspring was lower [50], or even changed over time: Weanling male descendants performed similarly, while in young adulthood high fat diet offspring showed higher locomotor activity [40]. Maternal high fat diet associated with anxiety-like behavior in F1 generation [42,50,81] and in F2 females [53]. Differences in the findings on the maternal dietary challenge on locomotor and anxiety-like behavioral responses in offspring might be related to different factors, such as different animal models, different exposure windows to diet (periconception, gestation, lactation), as well as the age at which animals were tested. Different diet composition might have influenced the outcomes as well, particularly if confounded with maternal obesity. However, even in the absence of maternal obesity, such as in our study and those of Johnson et al. [50] and Speight et al. [51], different maternal diets produce a different impact on offspring locomotion and anxiety-like behavior. The overall decline in locomotor activity with ageing, as documented in our study from adolescence to old age, is a commonly observed ageing deficit in rodents [82,83]. However, age-dependent changes in locomotion and anxiety-like behavior seem not to be correlated [83].

A Morris water maze is a well-established test of hippocampal-dependent learning and memory in rodents. Western diets may induce neurophysiological changes, such as impaired glucoregulation, reduced levels of neurotrophins, changes in levels of excitatory and inhibitory amino acids and neurotransmitters, neuroinflammation, and alterations in the structural integrity of the blood-brain barrier that can directly or indirectly impair hippocampal-dependent learning and memory operations [54,84]. Data on the effects of maternal high fat diet on the acquisition of offspring in special learning tests are contradictory. One study reported no effect of F0 high fat diet on the spatial working memory of F2 mice offspring of either sex [54]. In another study, maternal high fat diet associated with the impaired acquisition of spatial memory in adolescent males, but not in adult offspring; while in probe trial the results were inconsistent [40,41]. In an Alzheimer mouse model, administration of irradiated diet (i.e., rich in AGEs) for 11 months associated with significantly poorer memory, higher hippocampal levels of insoluble amyloid-beta and AGEs compared to littermates fed an isocaloric diet [85]. Counter to these data, piglets [48,49], and male (but not female) mice offspring [50] exposed to a Western diet or maternal high fat diet, respectively, showed improved spatial learning and memory compared with their control counterparts. In our study, female AGE-RD F1 and F2 offspring presented better working memory compared with the CTRL females. Shorter time to find the platform on days 2-to-4 could not be attributed to a higher locomotor activity of AGE-RD vs. CTRL offspring: Old F1 and F2 females of both dietary groups moved in an open field test with similar velocity (*p* = 0.909, and *p* = 0.156, respectively). A sexually dimorphic effect upon Morris water maze performance has been reported in offspring of dams challenged with high fat diet [41,50]. Sexual dimorphism within the limbic system has been clearly recognized [86], thus sexual dimorphism in response to maternal dietary challenge upon hippocampal-dependent function in the brain is not unexpected. Mechanisms underlying maternal AGE-RD-induced sex-dependent changes in offspring remain to be elucidated. 

Cross-sectional studies in apparently healthy humans suggest that accumulation of AGEs associates with cognitive decline. In elderly subjects, higher serum concentrations of methylglyoxal (AGE precursor) associated with poorer cognition and faster cognitive decline [87,88,89]. However, dietary methylglyoxal is rapidly degraded during the digestion process in the intestine, thus it exerts no influence on circulating methylglyoxal levels [90]. Plasma methylglyoxal is of endogenous origin, produced in numerous metabolic pathways (e.g., anaerobic glycolysis, gluconeogenesis, glyceroneogenesis, ketone body or threonine metabolism, degradation of monosaccharides and glycated proteins) [91]. A small cross-sectional study in elderly subjects hypothesized that AGEs contribute to cognitive dysfunction in cerebrovascular disease, as CML expression in brain microvessels and cortical neurons related to clinical dementia [92]. In a prospective study, a higher baseline urinary pentosidine excretion associated with accelerated cognitive ageing [93]. However, both latter studies probably reflect an association of endogenous AGEs with cognitive impairment, not of AGEs of dietary origin. The evidence of association is not a confirmation of causality. On the other hand, absorbed dietary CML adducts or peptides cross the blood-brain barrier and accumulate in brains [28], although it is not known in which target structures or cell types. Elucidation of this issue is cumbersome, as neither chromatographic, nor immunohistochemical detection makes it possible to distinguish between exogenous and endogenous AGEs.

The novel object recognition task is the most frequently used method in assessing non-spatial object memory in rodents. In this task, representation of the object resides mainly in the perirhinal cortex, whereas the hippocampus encodes spatial context features [94]. Rodents naturally incline to approach and explore novel, non-threatening objects using multiple senses. Chance performance is represented by 50%. Thus, sample object memory strength is inferred from the preference of the rodent to explore the novel object over the familiar object during the test session. We revealed no significant impact of F0 diet on the preference of novel object in F1 and F2 offspring. To our knowledge, maternal diet-induced intergenerational effects on novel object recognition have not been reported yet. In accordance with our data, maternal high fat diet from pre-pregnancy to the end of lactation did not affect the preference for the novelty of offspring of either sex [42]. On the other hand, lactational exposure of dams to cafeteria diet resulted in impaired performance of their weanling offspring in short time (5 min and 30 min after familiarization trail), while after 60 min, neither control, nor cafeteria diet offspring showed preference of a novel object [95]. In the cafeteria group, dopamine metabolism in the prefrontal cortex was significantly reduced, whereas serotonin metabolism was increased. Lower preference of a novel object has also been reported in weanling rats of dams administered cafeteria diet from weaning to weaning of their offspring; while in young adulthood, both groups of offspring performed similarly [47]. In this study, the test session was performed 24 h after the familiarization trial. In a different study, young adult offspring of dams exposed to cafeteria diet during lactation manifested intersexual differences in novel object recognition [45]. While control male offspring displayed intact memory 1 hour after initiation trial, maternal cafeteria diet delayed natural forgetting such that discrimination was also evident after 2 hours. In contrast, control females exhibited discrimination following both 1 and 2 hours, but the cafeteria diet offspring manifested impaired performance. The large variability of findings on the effects of maternal Western type diet on object recognition memory in offspring may be related to the sample session object exploration criterion and differences in the inter-session delay; different age of offspring subjected to testing, and presence of diet-induced maternal obesity [42,45,47] in some settings. Further studies are required to elucidate whether adiposity associated inflammation and other factors, rather than a maternal diet, per se, may explain the adverse effects found on offspring’s cognition. Available data suggest that nutritional programming of non-spatial object memory in rodents imposed by different Western type diets vary, might be age- and sex-dependent and be modulated if offspring are in latter stages of life fed control diet. 

Natural body-care behavior serves as an indicator of animal health. A “splash test” with a sucrose solution is generally used as an indirect measure of body care efficiency in rodents. However, in setting with sucrose, interpretation is not straightforward. Longer latency to groom or shorter grooming might be indicative of anxiety- or depressive-like behavior or anhedonia; while longer grooming might reflect higher self-care or higher sucrose preference. Whether self-grooming is indicative of maternal grooming of offspring remains unclear. Grooming, licking, nursing, and skin-to-skin contact are important mother-pup interactions, which might modulate early development and later behavioral phenotype of offspring. Two studies which followed maternal behavior showed that dams consuming cafeteria diet spent longer time licking, nursing, or feeding their offspring [46,51]. Interestingly, their offspring manifested delayed physical and neurobehavioral development [46], higher locomotor activity [51] and lower anxiety [46,51] compared with offspring of dams consuming control chow. In our experiment, control and AGE-RD dams presented similar behavioral phenotype, except for the outcomes of a splash test. This raises the question of whether offspring outcomes could have been affected by AGE-RD-induced changes in maternal care.

## 5. Conclusions

The main conclusion of the current study is that maternal intake of a diet with moderately increased AGEs content from puberty to weaning of offspring modulates manifestation of physiological reflexes, such as hindlimb placing, fore- and hind-limb grasp, negative geotaxis and surface righting. As well as a behavioral phenotype in later life, i.e., working memory in F1 and F2 female and anxiety-like behavior in F2 male offspring. It also suggests the possibility that the mechanism of AGE-RD action may differ between the sexes. Considering a rising preference of diets containing considerable amounts of AGEs particularly by a young generation, further studies on the effects and specific mechanisms of inheritance imposed by AGE-RD are required. It is of interest what the results would be if the AGE-RD F1 and F2 offspring would also consume AGR-RD. Potential heritability via the paternal line remains unclear.

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
