# Peer review of "Maternal Consumption of a Diet Rich in Maillard Reaction Products Accelerates Neurodevelopment in F1 and Sex-Dependently Affects Behavioral Phenotype in F2 Rat Offspring"

_foods, 2019, doi:10.3390/foods8050168_

Reviewer 1 Report

General comments

The study of maternal exposure to dietary AGEs is a very interesting topic that has not yet received enough attention.

The current study is very well designed and well written.

Since I am not an expert in behavioural and other tests on rats I am not in a position to judge the usefulness and the quality of the methods used.

I do, however, have some comments and recommendations for the authors.

Detailed comments

The AGE-rich diet was prepared by heating a standard diet at 120°C for 30 min. The authors claims that this AGE-RD was nutritionally equivalent to the standard diet, but do not present any data on the content of micronutrients of either diets. It is well known that thiamin, at least, is heat-sensitive and is probably partly degraded during the heat treatment. Some studies have compensated for this possible deficiency by adding supplements to the animals’ diet (see ref. 83 from the current manuscript). In the current study I strongly recommend measuring the most heat-sensitive micronutrients.

Neoformed compounds other than CML and CEL, such as acrylamide, furane and HMF should be quantified in the feeds.

The authors mention that AGE-modified proteins are less digestible, particularly if cross-lined (ref 19). This was not confirmed by Alamir et al. 2013, but if we consider that to be true, a low bioavaibility of proteins by the AGE-RD group could be responsible for the few effects observed among the offspring in the current study. A difference in the perinatal amount of protein in the diet has been described as having long term physiological effects. A protein digestibility effect cannot be excluded.

In the paragraph ‘2.2.1 Quantification of AGEs in feeds’ the authors should change the title since the Amadori product is not an AGE.

Then the authors explain that they used a conversion factor of 3.1 to calculate the concentration of the Amadori product from the initial quantification of furosine. However this calculation was not used subsequently in the presentation of the results. Data are still expressed as mg of furosine /100g of proteins.

I recommend not using the conversion factor (which may vary form one laboratory to the other) and keeping the data expressed as they are (i.e. mg furosine/100g).

Line 310 to 318: the authors described the exposure to CML and CEL but not furosine. Why?

Table 3 shows the % difference in AUC of presence of reflex manifestation. How can we compare the data? Can’t the authors apply a statistical test to reveal potential significant differences between the four groups (F1 and F2 male and female)?

The discussion of the current manuscript is long and useful although it is mainly presented as a review of the literature. It must be said that this study is only observational, that most of the tests performed show no effect of the F0 exposure to AGEs and that, in conclusion, many results cannot be discussed.

The authors should be more precise when they describe previous studies. For instance when only CML has been studied previously the authors should not describe it as an exposure to AGEs, but precisely as an exposure to CML (ex: line 563).

The authors should also be more precise both in the conclusion and the abstract. For instance, the sentence ‘…modulates manifestation of certain physiological reflexes and behavioral phenotype…’ should be much more precise. What is behind the word CERTAIN ? Since there are few modulations observed in this study the authors should list them in both the conclusion and the abstract.

Other comments

Please check the copy and paste error in the abstract

The word ‘rat’ should be present in the title of the manuscript.

Define AGE-RD as soon as it appears in the text.

The term  F(2, 82) and so one are not well define in the text.

Line 531: is it 25%-25% or 75%-25%

Author Response

We would like to thank the reviewer for spending her/his valuable time to review our manuscript, for constructive criticism, and for suggestions on how to improve our manuscript.

The AGE-rich diet was prepared by heating a standard diet at 120°C for 30 min. The authors claims that this AGE-RD was nutritionally equivalent to the standard diet, but do not present any data on the content of micronutrients of either diets. It is well known that thiamin, at least, is heat-sensitive and is probably partly degraded during the heat treatment. Some studies have compensated for this possible deficiency by adding supplements to the animals’ diet (see ref. 83 from the current manuscript). In the current study I strongly recommend measuring the most heat-sensitive micronutrients.

Published papers, (i.e., Diamanti-Kandarakis et al, J Mol Med (Berl) 2007, 85, 1413-20, Peppa et al., Diabetes 2003, 52, 2805-13) using 30 min. long exposure to 125oC to induce rise in AGEs in feed (i.e., a slightly higher temperature than that used in our study) report no heat-induced changes in macro- and micro-nutrients content, including vitamins. Thus, we anticipated that similar heating of chow could not affect its nutrient composition differently from published data, and thus that both diets are nutritionally equivalent. We added this statement into the manuscript (L 124-5).

Detailed description of composition of commercially available Ssniff diets is freely accessible on the internet. Therefore, we did not consider copying these data into our paper.

In the study mentioned by the reviewer (of Lubitz et al.), AGE-rich diet was prepared by irradiation. Details on which micronutrients were degraded by irradiation and to what extent are not given. There is no comparative study on the effect of mild heating vs. irradiation of feed on production of AGEs or effect on micronutrients. We might suppose that the effects differ profoundly. In experimental studies, irradiation is an unusual method to produce AGE-rich feed. In contrast to heating, the relevance of consumption of irradiated diet by humans is with high probability negligible.

Neoformed compounds other than CML and CEL, such as acrylamide, furane and HMF should be quantified in the feeds.

The authors agree with reviewer’s comment: the Maillard reaction includes the formation of a bewildering array of molecules, where furan, HMF, and acrylamide can be among the potentially most. toxic ones. Anyway, in this study we are interested in the potential intergenerational effects of consumption of diet rich in AGEs on development of offspring. Furan, HMF, and acrylamide are classified as potential carcinogens. We did not aim to study the effects of dietary Maillard reaction products on carcinogenesis. By the way, each sacrificed animal needs to be examined in detail. We did not detect tumors in the animals.

The authors mention that AGE-modified proteins are less digestible, particularly if cross-lined (ref 19). This was not confirmed by Alamir et al. 2013, but if we consider that to be true, a low bioavaibility of proteins by the AGE-RD group could be responsible for the few effects observed among the offspring in the current study. A difference in the perinatal amount of protein in the diet has been described as having long term physiological effects. A protein digestibility effect cannot be excluded.

We agree with the reviewer that data on bioavailability of AGE-modified proteins are contradictory. As mentioned - in rats, lower digestibility of extruded chow was not confirmed; while a clinical study suggests that the consumption of a diet rich in Maillard reaction products negatively affects protein digestibility (Seiquer et al., Am J Clin Nutr 2006; 83:1082-8). We tackle this issue in the revised paper (L78-82).

We did not analyze the content of cross-linking AGEs in administered feed, thus it is impossible even to estimate whether employed mild heating of chow could induce cross-linking of proteins relevant for nutritional status of the animals. Blockage of 1% vs. 2% lysine might not induce nutritional deficit. Neither clinical, nor experimental data indicate that habitual consumption AGEs-rich diet would result in malnutrition. Body weight curves of dams, data on offspring weight, and indicators of fat depots do not support the assumption of nutritional deficit.

In the paragraph ‘2.2.1 Quantification of AGEs in feeds’ the authors should change the title since the Amadori product is not an AGE.

As suggested, we changed the title into “Maillard reaction products”

Then the authors explain that they used a conversion factor of 3.1 to calculate the concentration of the Amadori product from the initial quantification of furosine. However this calculation was not used subsequently in the presentation of the results. Data are still expressed as mg of furosine /100g of proteins.

I recommend not using the conversion factor (which may vary form one laboratory to the other) and keeping the data expressed as they are (i.e. mg furosine/100g).

Lysine Amadori compound quantification was achieved through furosine quantification and even if we are aware about the formation of Amadori compound occurs during the early stage of the Maillard reaction, furosine offers a direct relationship of nutritionally blocked lysine. Along with the % of blocked lysine we decided to further comment the relationship between pathophysiological outcomes and CML and CEL. According to Krause et al. (Eur Food Res Technol 2003; 216:277-83), the concentration of hydrochloric acid is the most relevant factor that influences the formation of furosine stating from lysine Amadori compound. For this reason, 3.1 was used.

Line 310 to 318: the authors described the exposure to CML and CEL but not furosine. Why?

Furosine (lysine Amadori compound) was included in the % of nutritionally blocked lysine according to Nursten (see L 318-25).

Table 3 shows the % difference in AUC of presence of reflex manifestation. How can we compare the data? Can’t the authors apply a statistical test to reveal potential significant differences between the four groups (F1 and F2 male and female)?

In the revised paper we describe in more detail (L308-11) the approach to evaluate manifestation of those reflexes which show high between-day variability (as visible in supplementary file Fig. S5-S7). High variability is a well-known problem in behavioral studies on rodents. Daily comparison of the prevalence of presenting vs. not presenting animals (as indicated by asterisks in Fig. S5-S7) brings reliable statistical data but might not reliably reflected developmental trend. On the other hand, even without detectable between-group significance on single days throughout the study, a shift between the curves of prevalence might be clearly visible, indicating a delay of manifestation in one vs. the other group. Thus, as other authors, we used the area under the curve of daily prevalence in manifestation within the whole group as an estimation of potential difference in manifestation of reflexes. Since each group is characterized by a single AUC curve, percentual difference in AUC considered significant is estimated arbitrarily.

The discussion of the current manuscript is long and useful although it is mainly presented as a review of the literature. It must be said that this study is only observational, that most of the tests performed show no effect of the F0 exposure to AGEs and that, in conclusion, many results cannot be discussed.

The concept of “The developmental origin of health and disease” considers pre- and peri-natal factors, such as maternal nutrition, as crucial determinants of susceptibility to manifest chronic degenerative diseases in later life. Current experimental studies on maternal “unhealthy“ dietary cues focus on the effects of high fat diet (poor in AGEs due to low content of proteins), or cafeteria diet. Both approaches are generally confounded by maternal obesity, which might impose a greater influence on offspring development than the maternal diet itself. We believe that it is crucial to discuss the differences between the effects of maternal AGEs-rich diet and those of high-fat or cafeteria diets. Despite that a straightforward implication of our data to human situation is impossible, the emerging problem is that some tests in our study showed an effect, not that most of the test performed showed no effect of the exposure of the F0 generation: the effects were manifested under an increase in maternal dietary AGEs load realistically mimicking potential difference reached in humans consuming AGEs-rich vs. AGEs-poor diets, in two generations of offspring consuming only a standard diet.

As a lecturer in the course “Basics of medical research” I believe that our study is not an observational study but an experimental one. An experiment is a controlled study in which the researcher attempts to understand cause-and-effect relationships. The study is "controlled" in the sense that the researcher controls for the assignment of subjects (animals) into groups and for which intervention each group receives. In the analysis phase, the researcher compares group scores (dependent variables), and draws conclusions about whether the intervention (independent variable) had a causal effect on the dependent variable.

The authors should be more precise when they describe previous studies. For instance when only CML has been studied previously the authors should not describe it as an exposure to AGEs, but precisely as an exposure to CML (ex: line 563).

As suggested, in the revised paper we refer to CML (L 598)..

The authors should also be more precise both in the conclusion and the abstract. For instance, the sentence ‘…modulates manifestation of certain physiological reflexes and behavioral phenotype…’ should be much more precise. What is behind the word CERTAIN ? Since there are few modulations observed in this study the authors should list them in both the conclusion and the abstract.

In the abstract, concrete observed behavioral changes were listed. As suggested by the reviewer, we mention which reflexes showed different manifestation and which features of behavioral phenotype differed in offspring in conclusions (L724-6).

Other comments

Please check the copy and paste error in the abstract

We apologize, duplicity has been solved in the revised paper.

The word ‘rat’ should be present in the title of the manuscript.

As suggested, it had been included in the title

Define AGE-RD as soon as it appears in the text.

We apologize for this mistake; the abbreviation has been defined at its first mentioning (L 105).

The term  F(2, 82) and so one are not well define in the text.

We used repeated measures GLM for statistical evaluation of behavioral data. In results, we refer to ageing-associated differences, using a standard way of reporting repeated measures ANOVA, e.g. F(df, dferror)=number, p value. We apologize, but we are not sure what should we define better in terms used. If GLM indicated significance, potential significance of F0 diet was further discussed.

Line 531: is it 25%-25% or 75%-25% ?

We apologize for the misprint, ratio has been corrected to 75%:25% (L 564).

Reviewer 2 Report

Dear Authors,

It is a very well executed study. AGEs are deleterious to human health. There have been several studies towards detection of AGEs including ELISA. I appreciate if authors can support this rather than reporting as "flaw" concept.

--------

Due to the difference in raw 507 materials and composition of standard rodent diets, it is expectable that similar heat-processing will 508 yield different degree of AGE modification. In majority of studies, content of AGEs in diets has been 509 measured using unspecific ELISA method [63-64]. ELISA-based data on food AGEs content are 510 flawed: epitopes reacting with the employed antibody are not limited to AGE-moieties, while the 511 antibodies probably do not recognize all AGE epitopes [65-66]. The ELISA method quotes AGE 512 content of feed in kU/weight, not allowing for comparison with amounts of chemically defined AGEs, 513 as quantified in our study. 514
CML is a widely used marker to assess AGE-modification in foods

-------

on the whole it is a good study.

Author Response

We would like to thank the reviewer for devoting her/his time to help us to improve our paper.

As required by the reviewer, we modified the statement (L538-43).

Round  2

Reviewer 1 Report

About the Amadori products:

All the amounts of Maillard products in feeds are expressed in mg/100g protein, except for the Amadori products which are expressed in mg/kg (kg of ?).

Amadori should be expressed also in mg/100g protein.

Author Response

I appologize for typing in the data incorrectly. In the revised version I changed the content of lysine Amadori compound in both diets to amounts in mg/100g proteins.